# Blood Iodine as a Potential Marker of the Risk of Cancer in BRCA1 Carriers

**DOI:** 10.3390/nu16111788

**Published:** 2024-06-06

**Authors:** Adam Kiljańczyk, Milena Matuszczak, Wojciech Marciniak, Róża Derkacz, Klaudia Stempa, Piotr Baszuk, Marta Bryśkiewicz, Cezary Cybulski, Tadeusz Dębniak, Jacek Gronwald, Tomasz Huzarski, Marcin R. Lener, Anna Jakubowska, Angela Cheriyan, Marek Szwiec, Małgorzata Stawicka-Niełacna, Dariusz Godlewski, Artur Prusaczyk, Andrzej Jasiewicz, Tomasz Kluz, Joanna Tomiczek-Szwiec, Ewa Kilar-Kobierzycka, Monika Siołek, Rafał Wiśniowski, Renata Posmyk, Joanna Jarkiewicz-Tretyn, Ping Sun, Rodney J. Scott, Steven A. Narod, Jan Lubiński

**Affiliations:** 1Department of Genetics and Pathology, International Hereditary Cancer Center, Pomeranian Medical University, ul. Unii Lubelskiej 1, 71-252 Szczecin, Poland; adam.kiljanczyk@pum.edu.pl (A.K.); milena.matuszczak@pum.edu.pl (M.M.);; 2Read-Gene, Grzepnica, ul. Alabastrowa 8, 72-003 Dobra, Poland; 3Department of Clinical Genetics and Pathology, University of Zielona Góra, ul. Zyty 28, 65-046 Zielona Góra, Poland; 4Women’s College Research Institute, Women’s College Hospital, University of Toronto, Toronto, ON M5G 1N8, Canada; 5Department of Surgery and Oncology, University of Zielona Góra, Zyty 28, 65-046 Zielona Góra, Poland; 6OPEN, Kazimierza Wielkiego 24 St, 61-863 Poznań, Poland; 7Medical and Diagnostic Center, 08-110 Siedlce, Poland; 8Genetic Counseling Center, Subcarpatian Oncological Hospital, 18 Bielawskiego St, 36-200 Brzozów, Poland; 9Department of Gynecology, Gynecology Oncology and Obstetrics, Institute of Medical Sciences, Medical College of Rzeszow University, Rejtana 16c, 35-959 Rzeszow, Poland; 10Department of Histology, Department of Biology and Genetics, Faculty of Medicine, University of Opole, 45-040 Opole, Poland; 11Department of Oncology, District Specialist Hospital, Leśna 27-29 St, 58-100 Świdnica, Poland; 12Holycross Cancer Center, Artwińskiego 3 St, 25-734 Kielce, Poland; 13Regional Oncology Hospital, Wyzwolenia 18 St, 43-300 Bielsko Biała, Poland; 14Department of Clinical Genetics, Medical University of Bialystok, 15-089 Bialystok, Poland; 15Non-Public Health Care Centre, Cancer Genetics Laboratory, 87-100 Toruń, Poland; 16Medical Genetics, Hunter Medical Research Institute, Priority Research Centre for Cancer Research, Innovation and Translation, School of Biomedical Sciences and Pharmacy, Faculty of Health and Medicine, University of Newcastle, Pathology North, John Hunter Hospital, King and Auckland Streets, Newcastle, NSW 2300, Australia; rodney.scott@newcastle.edu.au

**Keywords:** microelements, cancer, BRCA1, breast cancer, ovarian cancer, nutrients

## Abstract

Breast cancer and ovarian cancer pose a significant risk for BRCA1 carriers, with limited risk-reduction strategies. While improved screening helps in the early detection of breast cancer, preventive measures remain elusive. Emerging evidence suggests a potential link between iodine levels and modulation of cancer risk, but comprehensive studies are scarce. We conducted a prospective study among 989 BRCA1 carriers to assess the association between blood iodine levels and breast and ovarian cancer risk. Using inductively coupled plasma mass spectrometry, we measured blood iodine levels and observed a negative association with breast cancer risk, with a significantly lower risk observed in quartile 4 (iodine > 38.0 µg/L) compared with quartile 1 (iodine < 30 µg/L) (HR = 0.49; 95%CI: 0.27–0.87; *p* = 0.01). Conversely, a suggestive increase in ovarian cancer risk was observed at higher iodine levels (HR = 1.91; 95%CI: 0.64–5.67; *p* = 0.25). No significant association was found between iodine levels and overall cancer risk. Our results suggest the potential of iodine to reduce breast cancer risk in BRCA1 carriers after prophylactic oophorectomy but require further validation and investigation of its effect on ovarian cancer risk and overall mortality. These findings highlight the need for personalized strategies to manage cancer risk in BRCA1 carriers.

## 1. Introduction

BRCA1 (MIM# 113705), a tumor suppressor gene located on chromosome 17q21, plays an important role in breast and ovarian cancer risk/development [1,2,3]. The lifetime risk of breast cancer in BRCA1 carriers is approximately 70%, and the risk of ovarian cancer is approximately 40% [4,5]. It is estimated that, in Poland, there are 200,000 carriers of BRCA1 mutation, and so far, we have identified over 10,000 in our clinics [6]. BRCA1 mutation genetic testing began in 1995, and since then, we have gained a lot of knowledge about cancer prevention. Enhanced screening of BRCA1 carriers with MRI surveillance leads to early detection of breast cancer but does not reduce the risk. Preventive bilateral mastectomy and salpingo-oophorectomy are crucial recommendations and can drastically reduce the risk and improve survival at the cost of adverse effects associated with these procedures [7,8,9,10,11,12,13]. Age, reproductive history, hormonal therapies, and oral contraception are modifiers of breast and ovarian cancer risk. Other modifiers of cancer risk in BRCA1 carriers include microelements, of which, to date, only serum arsenic and blood lead levels have been proven to influence the risk [14,15]. The high number of BRCA1 carriers in the population, coupled with the lifetime risk of ovarian and breast cancer, presents a need to identify additional factors that can reduce the risk.

Iodine is an essential element required for the proper functioning of the human body and is associated with thyroid function [16]. Radioactive iodine can increase the risk of developing several types of cancer [17,18,19]. Information on non-radioactive sources of iodine and cancer risk is scarce, although it has been proposed that iodine might protect against breast, ovarian, and endometrial cancer [20,21,22,23,24]. Notable quantities of iodine are found in seaweed, fish, iodized salt, eggs, dairy products, poultry, and beef liver. Our study aims to assess the association between blood iodine levels and the risks of breast and ovarian cancer in a prospective study of healthy BRCA1 carriers.

## 2. Materials and Methods

Our cohort was representative of cohorts of BRCA1 carriers from cancer family clinics generally characterized by an increased proportion of familial breast and ovarian cases. Participants of the study were 1119 initially unaffected adult women with a BRCA1 mutation, of which 130 had only baseline data collected and were lost to follow-up (88.38% completed at least one follow-up generally performed every 2 years). All carriers received genetic testing and counseling between 2011 and 2017 at the Clinical Hospitals of Pomeranian Medical University in Szczecin, Poland, or at an associated hospital or outpatient clinic. At recruitment and later every 6 months all carriers had surveillance, including breast MRI and ultrasound (USG), mammography, transvaginal USG, and blood CA125 analysis. During the first study visit, each subject had a fasting blood sample taken for genetic testing for BRCA1 mutations and later for elements measurement. A total of 10 mL of peripheral blood was taken for analysis from all subjects into a vacutainer tube containing ethylenediaminetetraacetic acid (EDTA). Each blood sample was drawn between 8 am and 2 pm and stored at −80 °C until analysis.

Usually, patients with the mutation are offered the possibility to participate in other studies. The deleterious BRCA1 variant warranted the inclusion of participants in the study. Written informed consent was given by all participants. Medical records were reviewed for diagnosis date, age at enrollment (blood draw), ovary removal (yes/no), smoking status (current, former, never), contraceptive use (ever/never), diabetes (yes/no), dietary supplement use (ever/never), hormonal therapy (ever/never), and BMI (low/normal/fat/obesity). The study adhered to the Helsinki Declaration and received approval from the Ethics Committee of Pomeranian Medical University in Szczecin, documented under the number KB-0012/73/10 on 21 June 2010.

### 2.1. Measurement of Blood Iodine Level

Blood samples were collected from fasting individuals via venipuncture using the Vacutainer^®^ System (product number 368381, Becton Dickinson, Plymouth, DEV, UK). The blood was then carefully aliquoted into new cryovials and frozen at −80 °C until it was analyzed.

The elemental composition of the samples was analyzed using inductively coupled plasma mass spectrometry (ICP-MS) with the NexION 350D instrument (PerkinElmer, Norfolk, VA, USA). Element determination was conducted in KED (Kinetic Energy Discrimination) mode, with rhodium serving as an internal standard to account for instrument drift and matrix effects. Specific parameters of the NexION 350D instrument used in the measurements can be provided upon request. For the analysis, blood samples were diluted 40-fold with a blank reagent (70 µL of blood to 2730 µL of buffer).

The blank reagent composed of high-purity water (>18 MΩ), TMAH (AlfaAesar, Kandel, Germany), Triton X-100 (PerkinElmer, Shelton, CT, USA), ethylenediaminetetraacetic acid (Merck, Darmstadt, Germany), and ethyl alcohol (Merck, Darmstadt, Germany). Calibration standards were created by diluting a 1000 µg/mL Iodine Standard stock solution (PerkinElmer Pure Plus, Shelton, CT, USA) with the blank reagent. The calibration method was matrix-matched, and the correlation coefficients for the calibration curve were always greater than 0.999.

The accuracy and precision of the measurements were assessed using certified reference materials (CRM): ClinChek^®^ Plasmonorm Whole Blood Level 1 (Recipe, Munich, Germany) and Seronorm Whole Blood Level 2 (Sero, Norway). Technical details, including mass spectrometer acquisition parameters and plasma operating settings, are available upon request. The testing laboratory also participates in the independent external quality assessment scheme, QMEQAS (Quebec Multielement External Quality Assessment Scheme), organized by the Institut National de Santé Publique du Québec.

### 2.2. Statistical Analysis

All study subjects were allocated to one of four categories (quartiles) depending on their blood iodine level. The cumulative risks of breast and ovarian cancer were calculated from the age at blood draw to the age of diagnosis of breast or ovarian cancer, death from another cause, or last follow-up. For estimating the risk of ovarian cancer, women with oophorectomy prior to blood draw were excluded, and subjects with oophorectomy in the follow-up period were censored at the time of oophorectomy. For the analysis of breast cancer risk, oophorectomy was included as a time-dependent variable. To estimate the ten-year cumulative risk of ovarian cancer, patients were followed from blood draw to date of preventive oophorectomy, ovarian cancer, 10 years of follow-up, last follow-up, or death from another cause. Univariate and multivariate Cox proportional hazards regression analyses were performed to estimate risk ratios (HRs) for cancer risk. The following variables were included in the multivariate models: iodine level (quartile), year of birth, age at blood draw (<40 years, 40–49.9 years, ≥50 years), oral contraceptive use (yes/no), hormone replacement therapy use (yes/no), smoking history (current, former never), and BMI (<18.5, 18.5–24.9, 25.0–29.9, ≥30.0). SAS, version 9.4, was used for all statistical analyses.

## 3. Results

The study group consisted of 989 women with a diagnosed BRCA1 mutation. The women were unaffected at the time of inclusion in the study. The mean age of enrollment (blood draw) was 44.0 years. The mean follow-up was 7.52 years, during which 173 new cancers were reported in various organs (122 breast cancer cases, 29 ovarian cancer cases, and 23 cancers at other sites). We excluded 84 patients from the analyses due to missing data. Group characteristics are presented in Table 1.

A total of 573 of the women had a risk-reducing oophorectomy at a mean age of 45.2 years. A total of 204 of the women had the oophorectomy prior to the blood draw, and 369 women had the oophorectomy in the follow-up period.

The distribution of iodine levels in the cohort is shown in Figure 1.

A significant negative association was observed between blood iodine levels and breast cancer risk (Table 2). Women with a blood iodine level greater than 38.0 µg/L had a significantly lower risk of breast cancer than women with blood iodine levels below 30 µg/L (quartile 4 vs. 1; HR = 0.49; 95%CI: 0.27–0.87; *p* = 0.01; Table 2). The crude and adjusted hazard ratios were similar. The mean iodine level was 33.22 µg/L for those who developed breast cancer and 35.40 µg/L for those who did not develop breast cancer (*p* = 0.004).

We observed an increased risk of ovarian cancer according to iodine levels in the BRCA1 carriers (Table 3). Those who had an iodine level above the median (34 µg/L) had a higher risk of ovarian cancer relative to those with a level below the median (crude HR = 2.15; 95%CI: 0.75–6.18; *p* = 0.16). This was not attenuated in the multivariate model (adjusted HR = 1.91; 95%CI: 0.64–5.67; *p* = 0.25). The mean iodine level was 50.56 µg/L for those who developed ovarian cancer and 34.71 µg/L for those who did not develop ovarian cancer (*p* = 0.25).

There were 240 women who had an iodine level below 34.0 µg/L (lowest quartile). In this quartile, there were 5 ovarian cancer cases and 33 breast cancer cases (38 cases in total). There were 240 women who had an iodine level above 38.0 µg/L (highest quartile). In this quartile, there were 11 ovarian cancer cases and 18 breast cancer cases (29 cases in total).

The results of the analysis between iodine level and the risk of any cancer are presented in Table 4. The association with iodine was not significant.

## 4. Discussion

In our study, we observed a significant association between a relatively high blood iodine level and a reduced risk of breast cancer (quartile 4 vs. quartile 1; HR = 0.49; 95%CI: 0.27–0.87; *p* = 0.01).

There was a suggestion of an increased risk of ovarian cancer associated with a high iodine level, but this was not statistically significant. Overall, the risk of all cancers combined was not associated with the serum iodine level.

Iodine deficiency as a cause of breast cancer has been a subject of interest. In Japan, a low incidence of breast cancer has been correlated with high consumption of dietary iodine, and adoption of a Western diet is associated with higher breast cancer risk [21,22,23,25,26]. Another study reported the effect of high dietary iodine intake on reducing the risk of breast, ovarian, and endometrial cancers [24]. However, only one previous study correlated serum iodine levels with breast cancer risk, performed in Sweden by Manjer et al. [20]. There was no association observed between iodine levels and breast cancer risk. A modest reduction in risk was found for the subgroup of women with high iodine and high selenium levels. However, to the best of our knowledge there is no prospective data about BRCA1 mutation carriers or the correlation between blood iodine level and cancer risk. A previous report from our center on selenium [27] and the aforementioned work, together with our results, are laying the groundwork for further studies of the effects of selenium and iodine on breast cancer risk, which we plan to examine in the future.

So far, most studies of iodine have been focused on its role in thyroid function. According to the WHO, a daily iodine intake of 150–299 µg is needed for the normal functioning of the thyroid [28]. In addition to the thyroid, other organs, such as breasts, the ciliary body of the eye, gastric mucosa, lacrimal and salivary glands, nervous system, ovaries, pancreas, placenta, prostate, skin, thymus, and uterus, also participate in iodine uptake [29]. Some researchers report that iodine may also have anti- and pro-inflammation functions [30,31,32]. Furthermore, a daily iodine intake of 1 mg or higher was linked to the antioxidant properties of this element [33,34]. Even higher doses of 1 to 6 mg of iodine intake a day were reported to have an improved outcome in fibrocystic breast disease at the cost of a high rate of side effects such as changes in the thyroid indices, acne, and short periods of increased pain [35].

There is a biological basis for the potential preventive effect of iodine on breast cancer. Iodine is crucial for physiological breast development—breast tissue contains iodine receptors (NIS), pendrin, and sodium-monocarboxylate transporter (SMCT). Precancerous lesions (breast dysplasia) can be caused in mice by iodine deficiency. Antioxidant, antiproliferative, and apoptotic effects in breast tissue are associated with iodine activity [23,33,36,37,38].

In studies on mice, it has been observed that iodine deficiency causes “hyperresponsiveness” to estradiol [37]. The higher sensitivity of BRCA1 carriers to iodine may be age-related, as the average age of breast cancer diagnosis is 45 years when there is high estradiol activity. In the study by Manjer et al., 25% of women were of age < 50 years, and in our cohort almost 70% were younger than 50 years [20].

Our study has several limitations. This is the first study of the effects of iodine on breast cancer risk in BRCA1 carriers, and the results should be validated in other populations. It may be that our data will not be relevant for other geographical regions and ethnic groups. Further studies should also consider interactions with other elements.

Additionally, our studies need to be extended to include the influence of iodine on all-cause mortality—there are suggestions that iodine deficiency is associated with more aggressive breast cancers [23,38]. It is also important to extend our findings of iodine levels on ovarian cancer risk to ensure that iodine supplements do not increase the risk of ovarian cancer. If confirmed, patients would have to adhere to different iodine diets depending on their mastectomy and oophorectomy status. At present, our data suggests that an iodine-rich diet or supplementary iodine may be indicated for women with a BRCA1 mutation who have had a preventive salpingo-oophorectomy and who have two breasts intact. We must also note that at this point, these results concern only BRCA1 carriers; further studies with a BRCA1-negative cohort are needed to assess whether we can apply the same correlation to a general population.

## 5. Conclusions

This is the first study to demonstrate an association between blood iodine levels in BRCA1 carriers and breast and ovarian cancer risk. The risk for breast cancer was decreased, but the risk for ovarian cancer presented a positive trend with increasing iodine levels. It is important to further study the association observed between iodine levels and cancer. The results of this suggest a potential for reduction in breast cancer risk in BRCA1 carriers using iodine but should only be considered after preventive oophorectomy.

## 6. Patents

A patent application (P.447790) has been submitted to the Patent Office of the Republic of Poland based on the results presented in the following communication.

## Figures and Tables

**Figure 1 nutrients-16-01788-f001:**
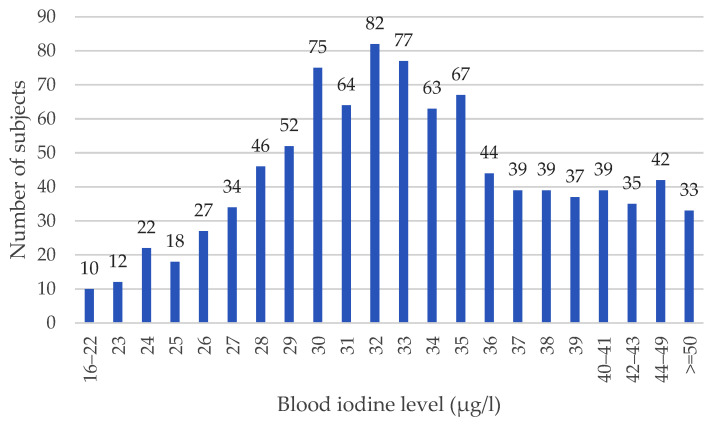
Distribution of iodine level in cohort of BRCA1 carriers (*n* = 989).

**Table 1 nutrients-16-01788-t001:** Group characteristics.

	Initially Unaffected (*n* = 989)
Age at enrollment (years) <50 ≥50 mean	775 (78.36%) 214 (21.64%) 44.0
Smoking Never Ever Missing data	720 (72.80%) 264 (26.69%) 5 (0.51%)
Hormonal therapy Never Ever Missing data	720 (72.80%) 263 (26.59%) 6 (0.61%)
Oophorectomy No Yes Missing data	413 (41.76%) 576 (58.24%) 0 (0.00%)
Oral contraceptive use Never Ever Missing data	501 (50.66%) 481 (48.64%) 7 (0.70%)
Diabetes No Yes Missing data	880 (88.98%) 62 (6.27%) 47 (4.75%)
Body Mass Index (kg/m^2^) <18.5 18.5–24.9 25.0–29.9 ≥30.0 Missing data	56 (5.66%) 553 (55.92%) 237 (23.96%) 95 (9.61%) 48 (4.85%)
Dietary supplements usage Never Ever Missing data	500 (50.56%) 489 (49.44%) 0 (0.00%)
New cancer diagnosis Yes No	172 (17.39%) 817 (82.61%)
New cancer site (*n* = 174) Breast Ovary Bladder Cervix Colon Kidney Leukemia Lung Pancreas Salivary gland Sarcoma Site unknown Skin Thyroid Urothelial Abdomen-CSU	122 (70.11%) 29 (16.67%) 2 (1.15%) 3 (1.72%) 2 (1.15%) 1 (0.57%) 2 (1.15%) 3 (1.72%) 1 (0.57%) 1 (0.57%) 1 (0.57%) 1 (0.57%) 1 (0.57%) 3 (1.72%) 1 (0.57%) 1 (0.57%)

**Table 2 nutrients-16-01788-t002:** Hazard ratio for breast cancer in relation to blood iodine level (quartile).

Variables	Breast Cases/ Total	Univariate HR (95%CI) P	Multivariate * HR (95%CI) P
Iodine level µg/L <30 30–34 34.1–38.0 >38.0 Total	33/240 31/237 32/240 18/240 114/957	1 0.86 (0.53–1.41) 0.56 0.90 (0.55–1.46) 0.65 0.52 (0.29–0.92) 0.03	1 0.89 (0.54–1.45) 0.63 0.90 (0.55–1.48) 0.68 0.49 (0.27–0.91) 0.01
Year of birth ≤1965 1965.01–1975 1975.01–1985 >1985	34/230 25/215 43/330 12/182	1 0.81 (0.48–1.35) 0.42 0.95 (0.61–1.50) 0.83 0.56 (0.29–1.08) 0.09	1 0.74 (0.31–2.05) 0.53 0.89 (0.25–3.22) 0.86 0.45 (0.11–2.01) 0.27
Age at blood draw ≤40 40.01–50 >50	59/549 26/208 29/200	1 1.12 (0.71–1.78) 0.62 1.24 (0.79–1.94) 0.34	1 1.62 (0.64–4.08) 0.31 1.53 (0.42–5.66) 0.52
Oophorectomy No Yes (time-dependent)	27/393 87/564	1 0.83 (0.57–1.21) 0.33	1 0.60 (0.36–0.99) 0.04
Oral contraceptive use No Yes	53/485 61/471	1 1.17 (0.81–1.69) 0.40	1 1.31 (0.88–1.94) 0.18
Hormone replacement therapy No Yes	83/697 31/259	1 0.87 (0.58–1.32) 0.51	1 0.83 (0.57–1.32) 0.43
Smoking No Current Former	54/539 33/215 27/203	1 1.60 (1.04–2.47) 0.03 1.33 (0.84–2.12) 0.22	1 1.59 (1.03–2.46) 0.04 1.31 (0.82–2.09) 0.26
BMI at blood taken ≤18.5 18.6–24.9 25.0–29.9 ≥30 Missing	10/54 65/541 27/232 9/93 3/37	1.54 (0.79–2.99) 0.21 1 0.95 (0.61–1.49) 0.83 0.86 (0.43–1.73) 0.67	1.75 (0.89–3.45) 0.11 1 0.92 (0.57–1.47) 0.73 0.83 (0.40–2.71) 0.61

* adjusted by all the variables listed in the left column.

**Table 3 nutrients-16-01788-t003:** Hazard ratio for ovarian cancer in relation to blood iodine level (quartile).

Variables	Ovarian Cases/ Total	Univariate HR (95%CI) P	Multivariate * HR (95%CI) P
Iodine level µg/L <30 30–34 34.1–38.0 >38 Total	5/188 1/188 9/189 11/189 26/754	1 0.18 (0.02–1.50) 0.12 1.69 (0.67–5.05) 0.35 2.15 (0.75–6.18) 0.16	1 0.17 (0.02–1.42) 0.10 1.47 (0.48–4.52) 0.50 1.91 (0.64–5.67) 0.25
Year of birth ≤1965 1965.01–1975 1975.01–1985 >1985	9/95 8/156 8/321 1/182	1 0.49 (0.20–1.22)0.13 0.25 (0.10–0.64)0.003 0.06 (0.01–0.50)0.006	1 1.01 (0.06–16.4) 1.00 0.42 (0.02–9.52) 0.59 0.09 (0.00–3.36) 0.19
Age at blood draw ≤40 40.01–50 >50	12/539 5/124 9/91	1 1.53 (0.55–4.23) 0.42 4.49 (1.99–10.1) 0.0003	1 0.57 (0.14–2.47) 0.45 1.09 (0.05–1.97) 0.62
Oral contraceptive use No Yes	16/359 10/394	1 0.54 (0.25–1.14) 0.10	1 0.80 (0.32–1.97) 0.62
Hormone replacement therapy No Yes	23/601 3/152	1 0.40 (0.12–1.32) 0.13	1 0.35 (0.11–1.25) 0.11
Smoking No Current Former	11/436 6/170 9/148	1 1.46 (0.58–3.71) 0.42 2.53 (1.09–5.85) 0.03	1 1.30 (0.48–3.56) 0.60 2.09 (0.84–5.21) 0.12
BMI at blood taken ≤18.5 18.6–24.9 25.0–29.9 ≥30 Missing	0/49 13/448 8/157 4/64 1/36	0 1 1.83 (0.76–4.42) 0.18 2.32 (0.76–7.13) 0.14	0 1 1.26 (0.50–3.17) 0.62 1.23 (0.37–4.02) 0.74

* adjusted by all the variables listed in the left column.

**Table 4 nutrients-16-01788-t004:** Hazard ratio for all cancers associated with blood iodine level.

Variables	All Cases/ Total	Univariate HR (95%CI) P	Multivariate * HR (95%CI) P
Iodine level µg/L <30 30–34 34.1–38.0 >38 Total	43/240 35/237 49/240 35/240 162/957	1 0.67 (0.43–1.05) 0.08 0.98 (0.65–1.48) 0.92 0.80 (0.51–1.25) 0.33	1 0.68 (0.44–1.07) 0.10 1.00 (0.66–1.52) 0.99 0.79 (0.51–1.25) 0.32
Year of birth ≤1965 1965.01–1975 1975.01–1985 >1985	54/230 39/215 54/330 15/182	1 0.84 (0.56–1.27) 0.40 0.83 (0.57–1.21) 0.32 0.62 (0.35–1.11) 0.11	1 1.40 (0.57–3.46) 0.46 1.81 (0.58–6.68) 0.31 1.16 (0.34–3.98) 0.81
Age at blood draw ≤40 40.01–50 >50	77/549 36/208 49/200	1 1.14 (0.77–1.70) 0.51 1.41 (0.98–2.02) 0.06	1 1.53 (0.73–3.22) 0.26 2.41 (0.77–7.70) 0.13
Oral contraceptive use No Yes	83/485 79/471	1 0.92 (0.68–1.26) 0.61	1 1.01 (0.73–1.42) 0.94
Hormone replacement therapy No Yes	122/697 50/259	1 0.67 (0.47–0.96) 0.03	1 0.61 (0.41–0.89) 0.01
Smoking No Current Former	78/539 41/215 43/203	1 1.42 (0.98–2.08) 0.07 1.47 (1.01–2.14) 0.04	1 1.40 (0.95–2.06) 0.09 1.37 (0.93–2.00) 0.11
BMI at blood taken ≤18.5 18.6–24.9 25.0–29.9 ≥30 Missing	10/54 86/541 43/232 19/93 4/37	1.15 (0.60–2.22) 0.67 1 1.11 (0.77–1.60) 0.59 1.49 (0.90–2.44) 0.12	1.16 (0.59–2.25) 0.67 1 0.95 (0.64–1.40) 0.80 1.27 (0.76–2.13) 0.37

* adjusted by all the variables listed in the left column.

## Data Availability

The data presented in this study are available on request from the corresponding author due to privacy.

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
