# Peer review of "Blood Iodine as a Potential Marker of the Risk of Cancer in BRCA1 Carriers"

_nutrients, 2024, doi:10.3390/nu16111788_

Round 1
Reviewer 1 Report
Comments and Suggestions for Authors
Introduction:
In introduction the authors cite serum arsenic level modifiers of cancer risk. Other heavy metals and biomarkers have an association with risk cancer. Please explain all serum and blood modifiers of cancer risk or remove the only information about arsenic.
Materials and Methods
The selection of study subjects is clear, adult women with a BRCA1 mutation. As said by the authors "Typically, patients with a mutation are offered the opportunity to participate in other research studies". Please explain the Representativeness of the exposed and enrolled cohort.
The authors declare that the women were unaffected at the time of inclusion in the study but not explain how demonstrate that outcome of interest was not present at start of study.
Please insert section about assessment of outcome
Furthermore the authors could include information about adequacy of follow up of cohorts (% complete follow up, description provided of subjects lost to follow up)
Results
In the results the authors reported Hazard ratio for all cancers associated with serum iodine level and particularly for breast and ovarian cancer according to iodine level (quartile).
In all results they consider the variables BMI as median and not by classification of BMI. Please consider the category 18.5-24.9 (normal) as reference and reported the risk for the others categories.
Author Response
Introduction:
In introduction the authors cite serum arsenic level modifiers of cancer risk. Other heavy metals and biomarkers have an association with risk cancer. Please explain all serum and blood modifiers of cancer risk or remove the only information about arsenic. –
Introduction section has been corrected to avoid misinterpretation – The authors meant to put an emphasis on currently available data on correlation between micronutrients and cancer risk In BRCA1 carriers. During the revision process, a study investigating correlation between blood lead level and cancer risk in BRCA1 carriers was published. It was also added to the introduction.
Materials and Methods
The selection of study subjects is clear, adult women with a BRCA1 mutation. As said by the authors "Typically, patients with a mutation are offered the opportunity to participate in other research studies". Please explain the Representativeness of the exposed and enrolled cohort.
Section has been added to Materials and Methods. The enrolled cohort comprised of initially unaffected women with available blood sample and at least one follow-up which occurred at least one year after baseline. The exposed cohort is representative of BRCA1 carriers from cancer family clinics.
The authors declare that the women were unaffected at the time of inclusion in the study but not explain how demonstrate that outcome of interest was not present at start of study.
Please insert section about assessment of outcome
Unaffected women were under frequent surveillance (breast MRI and ultrasound, mammography, transvaginal ultrasound and CA125 marker). Section has been added to Materials and Methods.
Furthermore the authors could include information about adequacy of follow up of cohorts (% complete follow up, description provided of subjects lost to follow up)
Subjects lost to follow-up (130 persons) added with % of complete follow-ups.
Results
In the results the authors reported Hazard ratio for all cancers associated with serum iodine level and particularly for breast and ovarian cancer according to iodine level (quartile).
In all results they consider the variables BMI as median and not by classification of BMI. Please consider the category 18.5-24.9 (normal) as reference and reported the risk for the others categories.
BMI median changed to standard classification of BMI (<18.5; 18.5-24.9; 25.0-29.9; ≥30.0). There was a slight change of multivariate results.
Minor changes
Several typos corrected.
Full Names of 4 authors added in the contribution section to distinguish authors with the same initials
Funding section corrected.
Number and percentage mismatch in table 1 corrected.
Many parts of the article were paraphrased to avoid repetition.
Reviewer 2 Report
Comments and Suggestions for Authors
In my opinion, in order to state that iodine levels are associated with a greater or lesser risk of breast or ovarian cancer in healthy subjects BRCA1 carriers, it is absolutely not sufficient to evaluate the levels of a single parameter and associate it with clinical history of 989 patients. The study is truly risky in its claims which are not supported by scientific data, but only by statistical associations.
The study also totally lacks a nutrigenomic and nutrigenetic background... it is a clinical study involving very simple blood tests.
The study, moreover, is very poorly designed... perhaps it would have been appropriate to also compare blood sodium levels between patients who develop ovarian cancer or breast cancer without carrying the BRCA1 gene. Or by analyzing other types of cancer, within a different population than BRCA1 carriers.
A scientific study conducted in this way makes no sense.
Author Response
We believe that our studies are very valuable pilot. We are aware that at the current level of research much of the molecular basis of our study is still undiscovered. Obviously, further investigation is needed.
Changes
Introduction:
Section with recommendation of preventive procedures of bilateral mastectomy, sapingo-oophorectomy added, basic information about BRCA1 mutation and prevalence in Poland added.
Materials and Methods:
Additional information about BRCA1 cohort added:
-subjects lost to follow-up
-representativeness of our cohort
-surveillance methods
Results:
BMI median changed to standard classification of BMI (<18.5; 18.5-24.9; 25.0-29.9; ≥30.0). There was a slight change of multivariate results.
Discussion:
We added a section about future directions.
In the future we plan to study:
-iodine correlation with cancer risk in general population,
-iodine and selenium correlation with cancer risk in BRCA1 carriers.
-expand our current cohort to further examine ovarian cancer risk since our number of ovarian cancer patients is relatively low.
We also added a section about general iodine properties- which organs uptake iodine and known influence on some of them and daily iodine intake.
At this point we tried not to emphasize correlation between iodine and ovarian cancer risk and its implications since results were not statistically significant.
Minor changes
Several typos corrected.
Full Names of 4 authors added in the contribution section to distinguish authors with the same initials
Funding section corrected.
Number and percentage mismatch in table 1 corrected.
Many parts of the article were paraphrased to avoid repetition.
Reviewer 3 Report
Comments and Suggestions for Authors
The authors have conducted a prospective study among 989 BRCA1 carriers to assess the association between blood iodine levels and breast and ovarian cancer risk and suggested the potential of iodine to reduce breast cancer risk in BRCA1 carriers after prophylactic oophorectomy but require further validation and investigation of its effect on ovarian cancer risk and overall mortality. It is a meaningful investigation and aims to give diet guidance for the BRCA1 carriers to decrease cancer risk. However, a major revision is needed.
1. The introduction is way too short to cover the background to let the reader understand the necessary reason to do the investigation. The authors should include the BRCA1 mutation instruction and its related studies in the past.
2. Have the authors considered other confounders, such as iodine utilization in the body, like receptor expression, etc.
3. A group of population without BRCA1 mutation would be better to be included as a negative control for exploring the relationship between iodine level and cancer risks with or without BRCA1 mutation.
4. The discussion should be rewritten. The interpretation of results could be further developed to explore the broader implications of the findings. It would be beneficial to explore the possibilities and future directions arising from the findings. Are there any follow-up studies that could build upon this research? Are there implications for practice or policy that warrant discussion?
Comments on the Quality of English Language
Moderate revision is needed.
Author Response
The authors have conducted a prospective study among 989 BRCA1 carriers to assess the association between blood iodine levels and breast and ovarian cancer risk and suggested the potential of iodine to reduce breast cancer risk in BRCA1 carriers after prophylactic oophorectomy but require further validation and investigation of its effect on ovarian cancer risk and overall mortality. It is a meaningful investigation and aims to give diet guidance for the BRCA1 carriers to decrease cancer risk. However, a major revision is needed.
- The introduction is way too short to cover the background to let the reader understand the necessary reason to do the investigation. The authors should include the BRCA1 mutation instruction and its related studies in the past.
Section with recommendation of preventive procedures of bilateral mastectomy, sapingo-oophorectomy added, basic information about BRCA1 mutation and prevalence in Poland added.
- Have the authors considered other confounders, such as iodine utilization in the body, like receptor expression, etc. – no data collected
At the time of creation of baseline and follow-up questionnaires these factors were not taken into consideration, therefore we cannot include aforementioned confounders.
- A group of population without BRCA1 mutation would be better to be included as a negative control for exploring the relationship between iodine level and cancer risks with or without BRCA1 mutation.
It is a very good suggestion, however current timeframe does not allow us to add negative control. We plan to examine negative controls in the future.
- The discussion should be rewritten. The interpretation of results could be further developed to explore the broader implications of the findings. It would be beneficial to explore the possibilities and future directions arising from the findings. Are there any follow-up studies that could build upon this research? Are there implications for practice or policy that warrant discussion?
Thank you on your input in this matter.
We added a section about future directions.
In the future we plan to study:
-iodine correlation with cancer risk in general population,
-iodine and selenium correlation with cancer risk in BRCA1 carriers.
-expand our current cohort to further examine ovarian cancer risk since our number of ovarian cancer patients is relatively low.
We also added a section about general iodine properties- which organs uptake iodine and known influence on some of them and daily iodine intake.
At this point we tried not to emphasize correlation between iodine and ovarian cancer risk and its implications since results were not statistically significant.
Other changes
BMI median changed to standard classification of BMI (<18.5; 18.5-24.9; 25.0-29.9; ≥30.0). There was a slight change of multivariate results.
Minor changes
Several typos corrected.
Full Names of 4 authors added in the contribution section to distinguish authors with the same initials.
Funding section corrected.
Number and percentage mismatch in table 1 corrected.
Expanded Materials and Methods section to provide additional information about patient enrollment, follow-up and surveiilance.
Many parts of the article were paraphrased to avoid repetition.
Round 2
Reviewer 2 Report
Comments and Suggestions for Authors
While appreciating the authors' effort, the manuscript continues to present the same type of critical issues. It is not a nutrigenetics paper, the experimental design is incorrect and the data are not sufficient to establish that blood Iodine is a marker of the risk of cancer in BRCA1 carriers. Among other things, although the authors have included in the text that these studies are clearly preliminary, the title is misleading, given that it takes for granted that blood Iodine is a marker of the risk of cancer in BRCA1 carriers
Author Response
As suggested by Academic Editors, we have changed the title of the article to ‘Blood Iodine as a Potential Marker of the Risk of Cancer in BRCA1 Carriers’ to emphasise the preliminary results of our study. Please find attached the responses to the editor's suggested changes.

Reviewer 3 Report
Comments and Suggestions for Authors
The necessary revisions have been made.
Author Response
Thank you for reviewing our article. Academic editors' suggestions and response in attachment.
